

# Comparative study on the composition of four different varieties of garlic

Cun Chen[1], Jing Cai[2], Song-qing Liu[1], Guo-liang Qiu[3], Xiao-gang Wu[4], Wei Zhang[5], Cheng Chen[1], Wei-liang Qi[1], Yong Wu[1] and Zhi-bin Liu[3]

[1] College of Chemistry and Life Science, Chengdu Normal University, Chengdu, PR China
[2] West China School of Pharmacy, Sichuan University, Chengdu, PR China
[3] Key Laboratory of Bio-Resource and Eco-Environment of Ministry of Education, College of Life Sciences, Sichuan University, Chengdu, PR China
[4] Chengdu Institute of Biology, Chinese Academy of Sciences, Chengdu, PR China
[5] College of Bioengineering, Sichuan University of Science and Engineering, Zigong, PR China

## ABSTRACT

Garlic is used as a medicinal seasoning worldwide. The aim of this work was to compare four varieties of garlic: 'Taicangbaipi', 'Ershuizao', 'Hongqixing', and 'Single-clove'; among them, 'Ershuizao' and 'Hongqixing' are unique to the Sichuan Province of China. Firstly, soluble sugar, starch, and the protein content of the garlic were analysed. There was more soluble sugar in 'Single-clove', total starch in 'Hongqixing', and protein content in 'Ershuizao' relative to the other three varieties, respectively. Gas chromatography–mass spectrometry analysis showed that 'Ershuizao' and 'Hongqixing' contained high levels of 5-hydroxymethylfurfural, which has antitumor, antioxidant, and cytoprotective effects. Indeed, the extracts from these two types of garlic were more effective at inhibiting tumour growth than that from the others. Moreover, the sulphide content and antimicrobial effects of 'Ershuizao' and 'Hongqixing' garlic were also higher than those of the other two types of garlic. In addition, changes observed in the membrane permeability and protein leakage suggest that the antimicrobial activity of the 'Ershuizao' and 'Hongqixing' extracts may be due to the destruction of the structural integrity of the cell membranes, leading to cell death.

# INTRODUCTION

Quality is an important factor that is taken into account when people choose food. The quality of food includes appearance, texture, aroma, taste, nutritional value, chemical composition, and functional properties. Nowadays, consumers are becoming increasingly interested in health issues, and the quality of food has gradually become a major factor in purchasing decisions. Along with some functional ingredients such as antioxidants and anti-aging substances, sugar, starch, and protein are the main nutrients that people consider when choosing foods. Different varieties, harvesting methods, and regions can lead to different levels of quality and contents. It was reported that fruit quality, size and shape traits significantly differed among 10 historic USA tomato varieties

Corresponding author
Zhi-bin Liu, liuzhibin@scu.edu.cn

(*Cho & Kim, 2008*). Narváez-Cuenca found that macronutrient contents (protein, fat, soluble dietary fibre, and insoluble dietary fibre) varied widely among 113 genotypes of potatoes (*Park et al., 2010*). Zheng found varying sugar levels in different varieties of *Lycium barbarum* and the congeneric species of *L. chinense* from different regions (*Zheng et al., 2010*). Monti found differing organoleptic and nutritional properties among different varieties of peach; these characteristics are related to their chemical composition (sugars and organic and amino acids) (*Monti et al., 2016*).

The *Allium* genus contains more than 600 different species that are widely distributed throughout Europe, North America, North Africa, and Asia (*Ozturk et al., 2012*). Most *Allium* species possess characteristic aromas and are edible. Garlic (*Allium sativum*) is consumed as a seasoning worldwide, and has been used for its important medicinal properties for centuries (*Itakura et al., 2001*; *Martins, Petropoulos & Ferreira, 2016*; *Matsutomo, Stark & Hofmann, 2018*; *Mukthamba & Srinivasan, 2015*). Many components in garlic, including sulphur-free compounds, work together to provide various health benefits. A double-blind crossover study conducted in a group of 41 men with moderate hypercholesterolemia showed that dietary supplementation of old garlic extracts was beneficial to the lipid profile and blood pressure of patients with moderate hypercholesterolemia (*Steiner et al., 1996*). Hosono showed that a flavour component obtained from garlic, diallyl trisulfide (DATS), exhibits antitumor activity (*Hosono et al., 2005*). Moreover, Durak showed the antioxidative effects of garlic extract in 11 patients with atherosclerosis (*Durak et al., 2004*). Garlic is considered an alternative health food because of those effects of improving the immune system.

China ranks first among the world's garlic-producing countries, and garlic consumers in Sichuan Province of China have access to four varieties of garlic in the local agricultural market: 'Taicangbaipi', 'Ershuizao', 'Hongqixing', and 'Single-clove' among them, 'Ershuizao' and 'Hongqixing' are unique to Sichuan Province of China. The main objective of the present study was to analyse the nutritional value of these different types of garlic, and to compare the antimicrobial and antitumor effects of their extracts.

## MATERIALS AND METHODS

### Materials and preparation

Four different varieties of garlic were purchased from the local agricultural market; the garlic bulbs and peeled garlic cloves are shown in Fig. 1. 'Taicangbaipi' and 'Ershuizao' were purchased in supermarket of Chengdu, and 'Hongqixing' and 'Single-clove' were bought in local market of Wenjiang. The garlic bulbs were peeled and crushed.

### Determination of soluble sugar and starch

Garlic juices (0.3 g) were extracted in 15 mL of 80% ethanol, and then centrifuged at 7,000 rpm for 10 min; the supernatant was retained for soluble sugar determination, whereas the precipitate was kept for starch extraction (*Irigoyen, Einerich & Sanchez-Diaz, 1992*). The precipitate was dissolved in 12 mL of 1.1% HCl and thoroughly mixed. The solution was heated in a water bath at 100 °C for 30 min to extract the starch, and

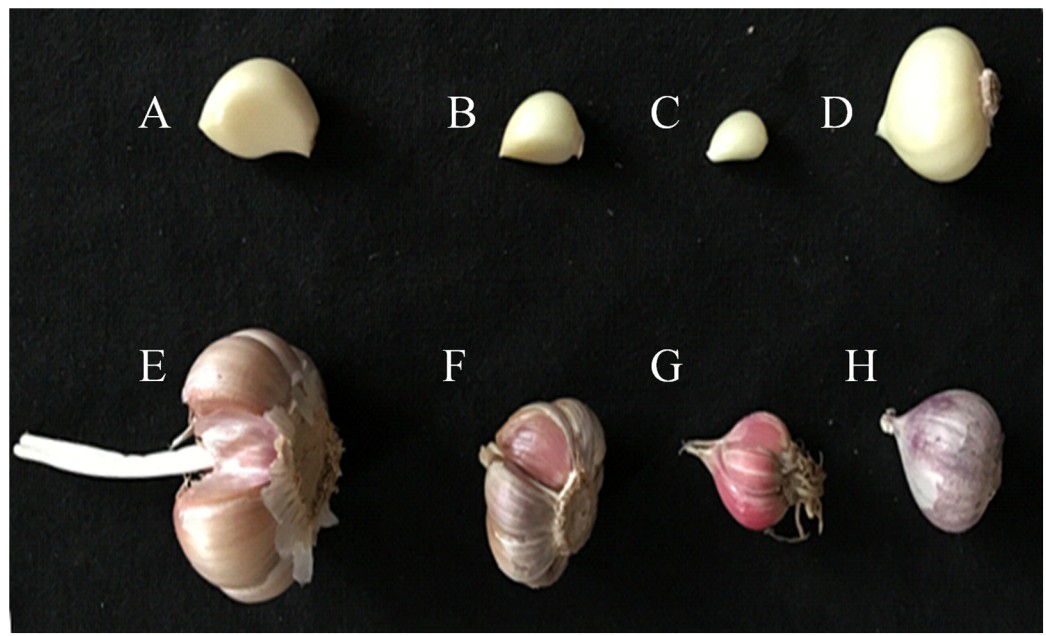

**Figure 1 Four different varieties of garlic.** (A–D) Peeled garlic cloves; (E–H) Garlic bulbs. (A, E) Taicangbaipi; (B, F) Hongqixing; (C, G) Ershuizao; (D, H) Single-clove.

after cooling, it was centrifuged at 7,000 rpm for 10 min; the supernatant was retained and diluted to 50% solutions.

The total soluble sugar and starch were estimated using anthrone reagent (*Sadasivam & Manickam, 1997*). One millilitre of the diluted solutions were added to five mL of freshly prepared anthrone reagent (0.2 g anthrone was dissolved in 100 mL of 72% sulphuric acid), and the mixtures were heated in a boiling water bath for 10 min. The tubes were removed and cooled, and then the absorbance of the content was measured at 625 nm in a spectrophotometer (Beijing Purkinje General Instrument Co., Ltd., Beijing, China). The amount of total sugar present in the sample was calculated from a standard curve drawn from variable amounts of glucose and total starch obtained from a standard curve for potato starch.

## Determination of protein content

Garlic juices (one g) were extracted in 3.5 mL protein extraction buffer containing 15% (v/v) 1M Tris–HCl (pH 8), 25% (v/v) glycerol, and 2% (w/v) polyvinylpyrrolidone. The mixture was kept on ice for 3 h, and then centrifuged at 10,000 rpm for 20 min at 4 °C. Samples of each supernatant (200 μL) were treated with 800 μL Coomassie Brilliant Blue G-250. After 10 min, the mixtures were subjected to colorimetric analysis in a T6-spectrophotometer (Beijing Purkinje General Instrument Co., Ltd., Beijing, China) at 595 nm. A standard curve was drawn using bovine serum albumin.

## GC–MS analysis

Gas chromatography analyses were performed using a Shimadzu GC/MS-QP2010 system (Kyoto, Japan). A Rtx-Wax capillary column (30 m × 0.25 mm, 0.25 μm) was used, with

helium carrier gas flowing at 1 mL/min; the column temperature was first kept at 60 °C for 5 min, then increased by 10 °C/min up to 280 °C. The injection port temperature was maintained at 220 °C. The MS data libraries NIST05.LIB and NIST05s.LIB were used for the spectrum analyses and compound identification.

## Microbial strains, culture, and antimicrobial test

*Xanthomonas campestris* pv. *malvacearum* (BNCC138498), *Pseudomonas syringae* (BNCC134219), *Fusarium proliferatum* (BNCC143058), and *Alternaria brassicicola* (CICC264) were obtained from Beina Chuanglian Biological Research Institute (Beijing, China). The garlic juices were mixed with deionized water at pH 3.0 at a ratio of 1:2 g/mL (*Chen et al., 2017*), for 2 h at 4 °C.

Oxford cup assays were used in the antimicrobial experiments, with some modifications (*Shang et al., 2014*). Bacteria (*X. campestris* pv. *malvacearum*, *P. syringae*) were grown to an $OD_{600}$ of 0.6 in nutrition broth (NB) at 28 °C in a shaker, then 100 mL NB agar was mixed with one mL of the bacterial solutions and added to the plates. Fungi (*F. proliferatum* and *A. brassicicola*) were cultivated on potato dextrose agar (PDA) plates at 28 °C. The spores were washed with 10 mL of sterile water and centrifuged at 6,000 rpm for 5 min to remove the supernatant. The spores were then mixed with sterile water to a concentration of approximately $2 \times 10^7$/mL as assessed by a haemocytometer. Then, 100 mL PDA agar was mixed with 0.5 mL spore suspension and added to the plates. Four Oxford cups (six-mm diameter) were placed above the agar surface, and 20 μL garlic extract ('Taicangbaipi', 'Ershuizao', 'Hongqixing', and 'Single-clove') was added to each cup, respectively. The bacterial inhibition zones were observed after 24 h at 28 °C, and the fungal inhibition zones were observed after 2–4 days at 28 °C. Three replicates were performed for each treatment.

## Cell membrane permeability

After 24 h of incubation at 28 °C, *X. campestris* pv. *malvacearum* cells were treated with the extracts of the four garlic varieties. The culture samples (five mL) were treated with 500 μL of the garlic extracts, and the mixtures were incubated at 28 °C for 1–12 h, respectively. Conductance was recorded using a conductivity meter (DDS-307; SPSIC-Rex Instrument Factory, Shanghai, China) after incubation, and was expressed as the bacterial membrane permeability relative conductivity and measured according to the method of *Diao et al. (2014)*.

## Integrity of cell membrane

Bacterial cultures containing 500 μL of the extracts of the four garlic varieties were centrifuged for 10 min at 6,000 rpm, and the supernatants were obtained. The supernatants (200 μL) were treated with 800 μL Coomassie Brilliant Blue G-250, and the mixtures were analysed at 595 nm on a T6-spectrophotometer (Beijing Purkinje General In-strument Co., Ltd, Beijing, China). The breakdown of bacterial cell membrane integrity manifested as protein leakage.
**Table 1 Soluble sugar, total starch, and protein content in 'Taicangbaipi', 'Hongqixing', 'Ershuizao', and 'Single-clove' extracts.**

| Varieties of garlic | Soluble sugar (mg/g) | Total starch (mg/g) | Protein (mg/g) |
|---|---|---|---|
| Taicangbaipi | 32.31 ± 0.15* | 163.51 ± 0.41 | 21.09 ± 0.16 |
| Hongqixing | 34.14 ± 0.08 | 287.71 ± 0.46 | 45.23 ± 0.11 |
| Ershuizao | 80.47 ± 0.20 | 263.98 ± 0.33 | 51.04 ± 0.32 |
| Single-clove | 186.91 ± 0.45 | 205.66 ± 0.13 | 40.36 ± 0.25 |

**Note:**
* Values represent the means of three independent replicates.

## Cytotoxic effects (antitumor) of the extracts of different garlic varieties on cells

Cell Culture: Colorectal cancer (CRC) cells (SW480 and HCT116) were obtained from the American Type Culture Collection (Rockville, MD, USA). The extracts of the four garlic varieties were dissolved in phosphate-buffered saline. The dilution ratios were 0, 1/32, 1/16, 1/8, 1/4, and 1/2. SW480 and HCT116 cells were seeded in 96-well plates. Various concentrations of the garlic extracts were applied to the cells, and cell viability was analysed after 72 h of incubation.

# RESULTS

## Soluble sugar, total starch, and protein content in different varieties of garlic

As shown in Table 1, different varieties of garlic resulted in significantly different soluble sugar, total starch, and protein content. The highest soluble sugar content was observed in 'Single-clove', at a mean of 186.91 mg/g. The mean starch levels of 'Hongqixing' and 'Ershuizao' were 287.71 and 263.98 mg/g, respectively, which were higher than that of both 'Taicangbaipi' and 'Single-clove'. It was also observed that 'Hongqixing' and 'Ershuizao' had higher protein contents, with mean levels of 45.23 and 51.04 mg/g, respectively. However, the overall content of these parameters in the 'Taicangbaipi' extract was relatively lower than those in the other three extracts.

## Constituents by GC–MS analysis

The identities of the compounds in the four varieties of garlic are listed in Table 2, and the GC–MS spectra of the constituents is shown in Figs. 2 and 3. The most abundant constituent in the 'Taicangbaipi' extract was B-D-fructofuranosyl α-D-glucopyranoside (27.34%), followed successively by ethylic acid (22.40%) and 2-amino-5-methylbenzoic acid (17.05%). The main constituents of the 'Hongqixing' extract were B-D-fructofuranosyl α-D-glucopyranoside (44.18%) and 5-hydroxymethylfurfural (26.78%). The primary compounds in the 'Ershuizao' extract were 5-hydroxymethylfurfural (47.10%), and B-D-fructofuranosyl α-D-glucopyranoside (31.00%). Finally, the primary constituents of the 'Single-clove' extract were B-D-fructofuranosyl α-D-glucopyranoside

Table 2 Major composition of 'Taicangbaipi', 'Hongqixing', 'Ershuizao', and 'Single-clove'.

| Compound name | Taicangbaipi | | Hongqixing | | Ershuizao | | Single-clove | | Ret. index |
|---|---|---|---|---|---|---|---|---|---|
| | R.Time | Area% | R.Time | Area% | R.Time | Area% | R.Time | Area% | |
| Ethylic acid | 3.086 | 22.40 | – | – | – | – | – | – | 576 |
| Hydroxyacetone | 3.544 | 1.93 | 3.300 | 0.37 | 3.304 | 0.35 | 3.332 | 1.38 | 698 |
| Dimethylethylene glycol | – | – | – | – | 4.424 | 0.19 | – | – | 743 |
| 2-Amino-5-methylbenzoic acid | 6.982 | 17.05 | 6.817 | 0.59 | 6.831 | 5.96 | 6.846 | 9.07 | 1,575 |
| 2,4-Dihydroxy-2,5-dimethyl-3(2H)-furan-3-one | – | – | 8.472 | 0.16 | 8.491 | 0.28 | – | – | 1,173 |
| Larixic acid | – | – | 10.662 | 3.16 | 10.665 | 2.94 | – | – | 1,063 |
| Methyl 2-oxohexanoate | – | – | – | – | 11.483 | 0.09 | – | – | 1,020 |
| Levulinic acid | 11.708 | 0.58 | 11.667 | 0.79 | 11.683 | 0.82 | 11.692 | 0.65 | 1,011 |
| 3,5-Dihydroxy-6-methyl-2,3-dihydro-4H-pyran-4-one | 11.840 | 3.94 | 11.800 | 3.31 | 11.808 | 4.16 | 11.818 | 2.91 | 1,269 |
| 3-Vinyl-1,2-dithiacyclohex-4-ene | – | – | 12.520 | 0.98 | 12.583 | 0.16 | – | – | 1,134 |
| 3-Vinyl-1,2-dithiacyclohex-5-ene | 12.943 | 1.14 | 12.954 | 4.46 | 12.974 | 1.48 | – | – | 1,134 |
| 5-Hydrxoymethylfurfural | – | – | 13.234 | 26.78 | 13.261 | 47.10 | 13.456 | 4.88 | 1,163 |
| Acetoglyceride | – | – | 13.550 | 3.49 | – | – | – | – | 1,091 |
| Diallyl trisulfide (DATS) | 14.426 | 0.74 | 14.422 | 1.64 | – | – | 14.421 | 0.38 | 1,350 |
| Dodecanal | 15.923 | 0.37 | 15.915 | 0.36 | 15.918 | 0.37 | 15.922 | 0.29 | 1,402 |
| β-D-fructofuranosyl α-D-glucopyranoside | 16.626 | 27.34 | 16.686 | 44.81 | 16.702 | 31.00 | 16.674 | 61.07 | 1,444 |
| Cyclododecane | 16.825 | 2.36 | – | – | – | – | 16.817 | 1.97 | 1,439 |
| 3,5-Di-tert-butylphenol | – | – | – | – | 17.381 | 0.21 | 17.386 | 1.65 | 1,555 |
| Hexadecane | – | – | – | – | – | – | 18.434 | 0.24 | 1,612 |
| 3-Deoxy-d-mannoic lactone | – | – | 18.642 | 1.86 | | | 18.633 | 0.22 | 1,625 |
| 1,3-Dimethylbarbituric acid | – | – | 18.741 | 2.01 | – | – | – | – | 1,532 |
| 3,6-Diisobutyl-2,5-piperazinedione | 18.825 | 1.41 | 18.808 | 1.11 | – | – | – | – | 1,636 |
| Carbamic acid, N-methyl-N-[6-iodo-9-oxabicyclo[3.3.1] nonan-2-yl]-, ethyl ester | – | – | 18.917 | 0.23 | – | – | – | – | 1,922 |
| l-Alanine, N-butoxycarbonyl-, isobutyl ester | 19.528 | 5.05 | 19.548 | 1.44 | 19.413 | 0.11 | 19.502 | 7.20 | 1,619 |
| 8-Methylheptadecan | – | – | – | – | – | – | 19.667 | 0.60 | 1,746 |
| Methyl 9-methylheptadecanoate | – | – | – | – | – | – | 19.975 | 0.43 | 2,013 |
| Tetradecanoic acid | 20.381 | 0.84 | – | – | 20.374 | 0.19 | 20.376 | 0.17 | 1,769 |
| Benzenesulfonic acid butyl amide | – | – | – | – | – | – | 20.874 | 0.19 | 1,797 |
| Pentadecanoic acid | 21.358 | 0.29 | – | – | 21.492 | 0.09 | – | – | 1,869 |
| Hexadecanoic acid, methyl ester | 22.216 | 0.53 | 22.203 | 0.28 | 22.209 | 0.19 | 22.215 | 0.80 | 1,878 |
| 9-Hexadecenoic acid | 22.390 | 1.20 | | | 22.375 | 0.22 | 22.383 | 0.22 | 1,976 |
| Hexadecanoic acid | 22.584 | 5.33 | 22.570 | 1.15 | 22.574 | 1.41 | 22.577 | 2.26 | 1,968 |
| Dibutyl phthalate | – | – | – | – | 22.717 | 0.96 | – | – | 2,037 |
| Methyl (10E)-10-octadecenoate | – | – | – | – | – | – | 24.033 | 0.13 | 2,085 |
| Oleic acid | 24.401 | 5.05 | 24.387 | 0.72 | 24.390 | 0.98 | 24.399 | 1.89 | 2,175 |
| Octadecanoic acid | 24.594 | 2.44 | | | 24.583 | 0.74 | 24.592 | 1.42 | 2,167 |
| 2-Ethyl-5-tridecylpyrrolidine | – | – | 26.733 | 0.28 | – | – | – | – | 2,160 |

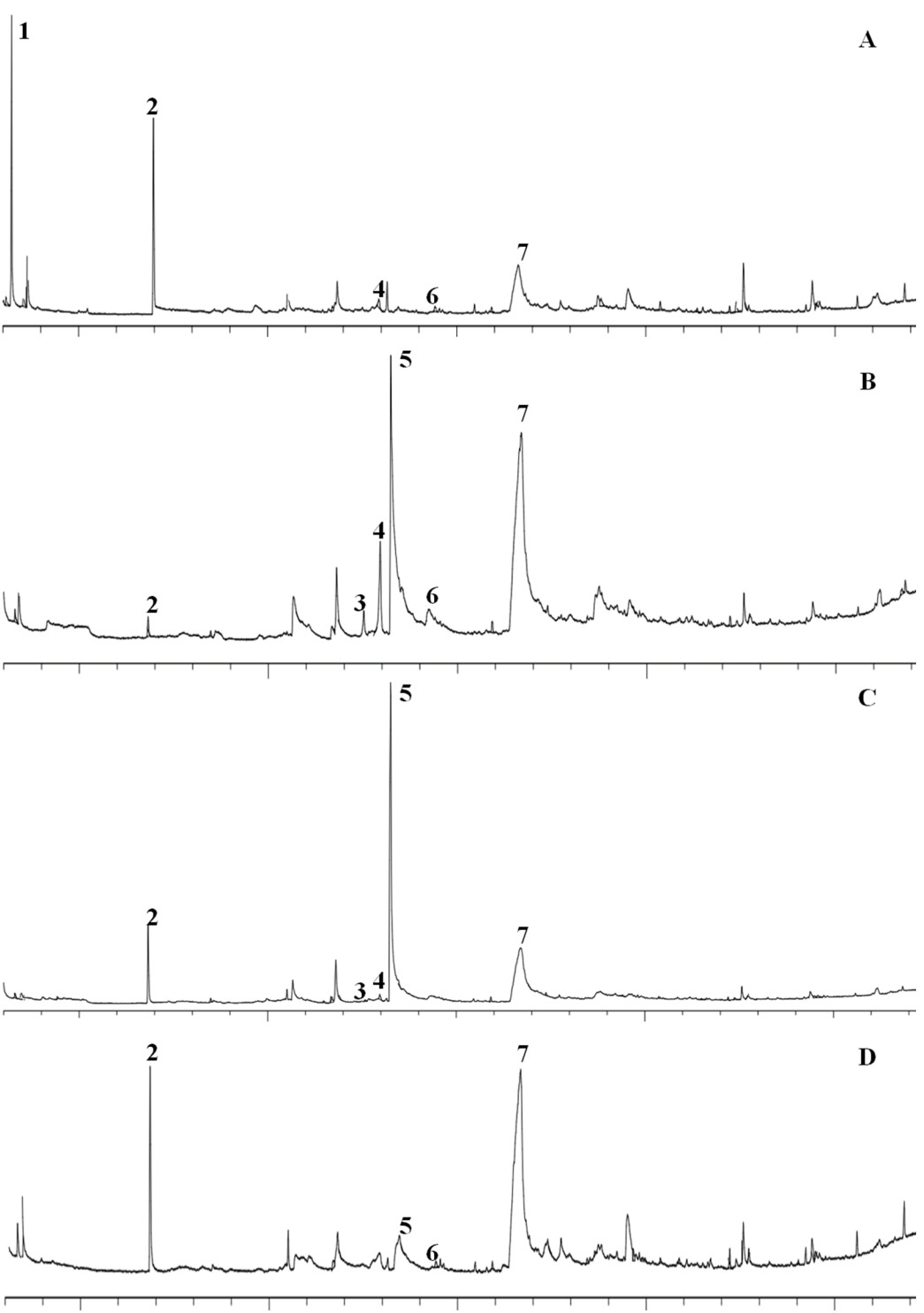

**Figure 2 GC–MS spectra of the constituents of four different varieties of garlic.** (A) Taicangbaipi; (B) Hongqixing; (C) Ershuizao; (D) Single-clove. Constituents: (1) ethylic acid; (2) 2-amino-5-methyl-benzoic acid; (3) 3-vinyl-1,2-dithiacyclohex-4-ene; (4) 3-vinyl-1,2-dithiacyclohex-5-ene; (5) 5-hydr-xoymethylfurfural; (6) diallyl trisulfide (DATS); (7) β-D-fructofuranosyl α-D-glucopyranoside.

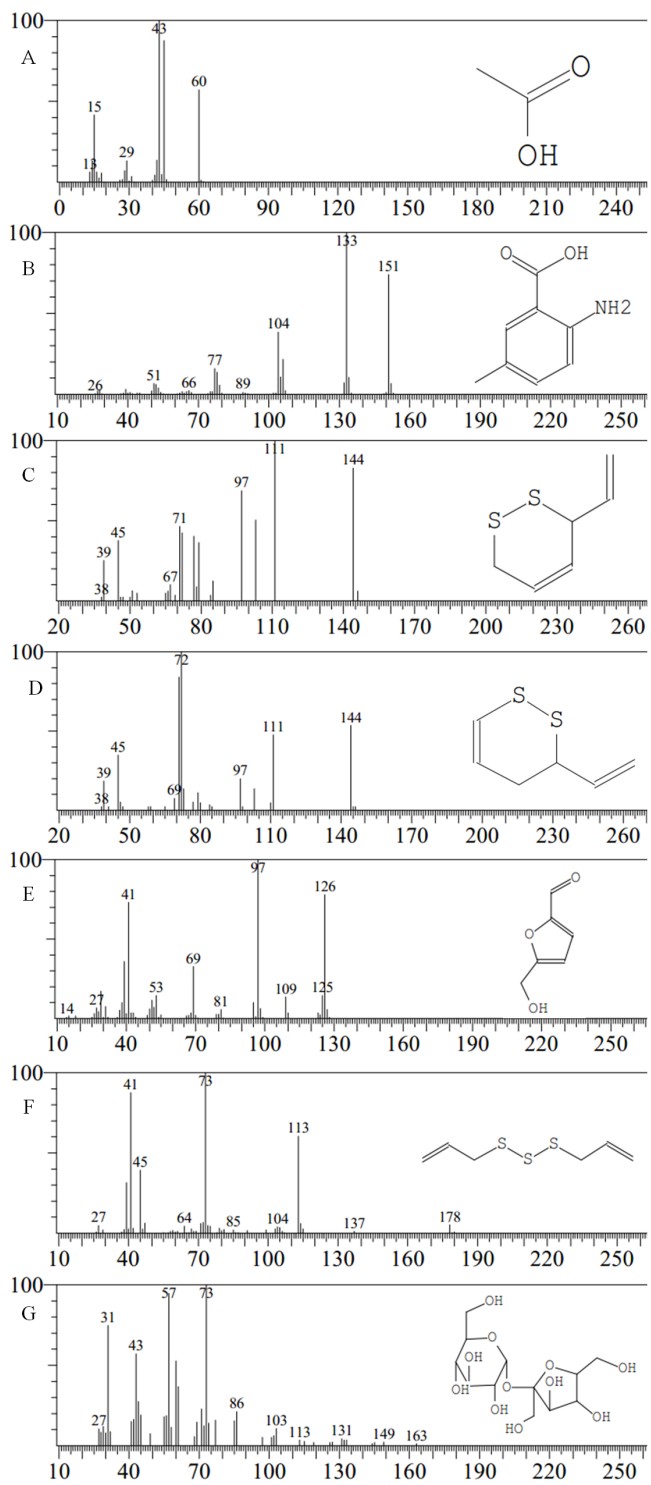

**Figure 3 Mass spectra of major compounds.** Constituents: (A) ethylic acid; (B) 2-amino-5-methyl-benzoic acid; (C) 3-vinyl-1,2-dithiacyclohex-4-ene; (D) 3-vinyl-1,2-dithiacyclohex-5-ene; (E) 5-hydr-xoymethylfurfural; (F) diallyl trisulfide (DATS); (G) β-D-fructofuranosyl α-D-glucopyranoside.

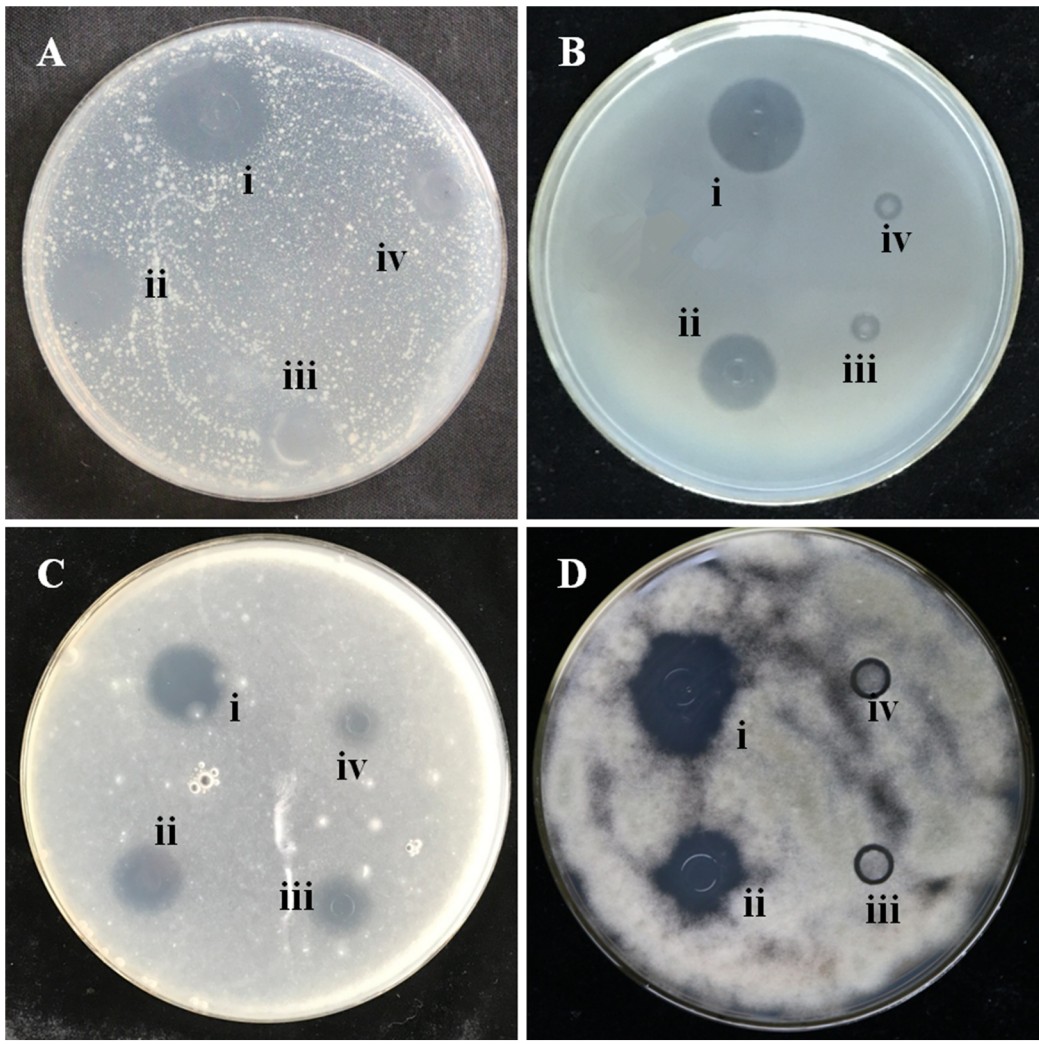

**Figure 4** **Zones of inhibition caused by the extracts of four varieties of garlic on seeded agar plates.** Each cup had 20 μL of garlic extract added to it (A) *X. campestris* pv. *Malvacearum*; (B) *P. syringae*; (C) *F. proliferatum*; (D) *A. brassicicola*. (i) Hongqixing; (ii) Ershuizao; (iii) Taicangbaipi; (iv) Single-clove.

(61.07%), 2-amino-5-methylbenzoic acid (9.07%), and 5-hydroxymethylfurfural (4.88%), in descending order.

## Activity of the extracts from different garlic varieties against plant pathogenic bacteria and fungi

In order to determine the antimicrobial effects of the extracts of the different garlic varieties, two types of bacteria (*X. campestris* pv. *malvacearum*, *P. syringae*) and two types of fungi (*F. proliferatum* and *A. brassicicola*) were treated with the four garlic extracts. After incubation, the zones of inhibition showed that the 'Hongqixing' extract had the highest activity against the microorganisms, followed by 'Ershuizao', 'Taicangbaipi', and 'Single-clove', in descending order of activity. The experimental results are shown in Fig. 4.

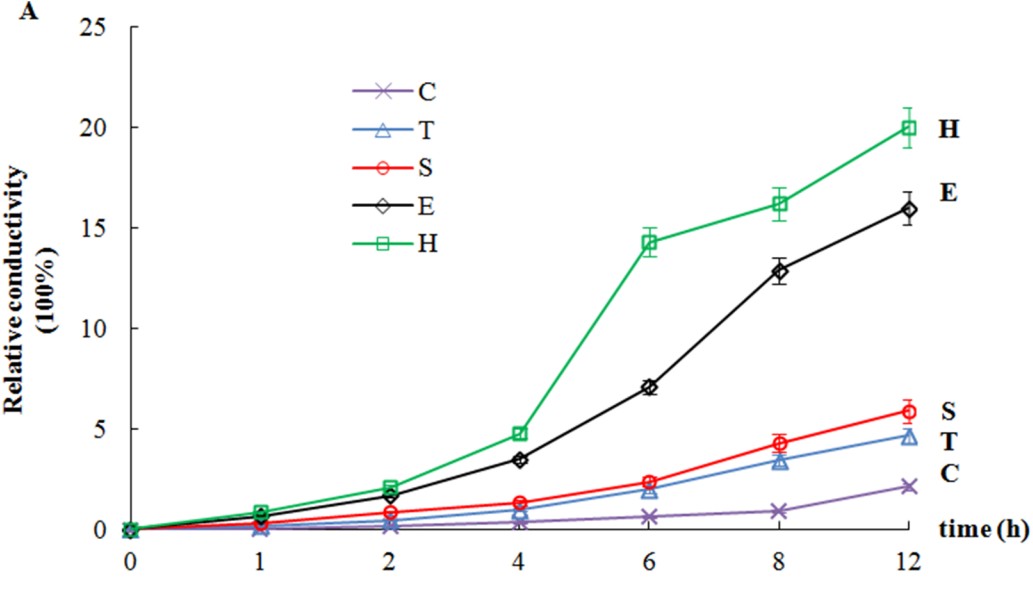

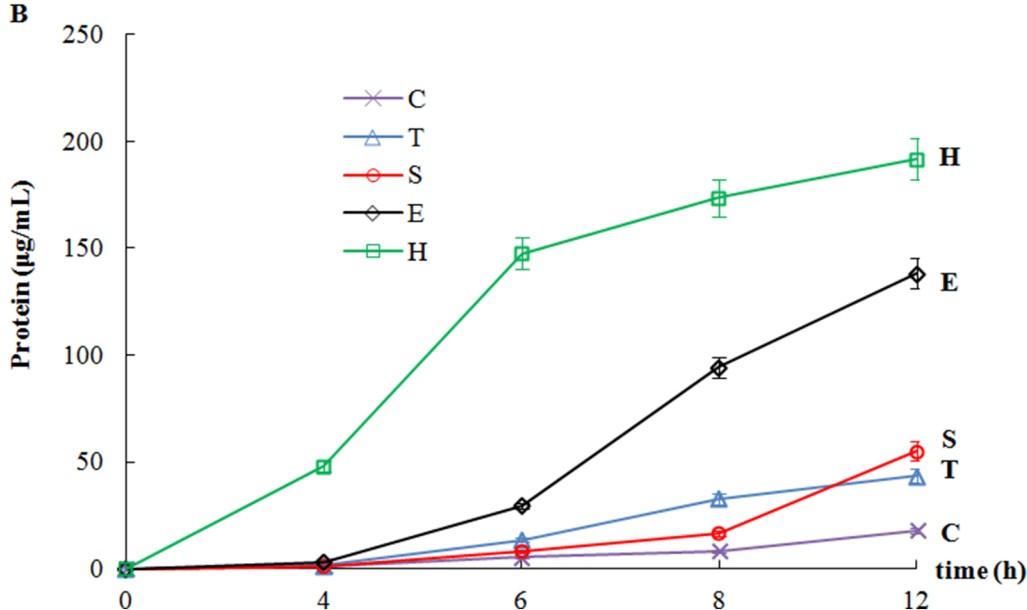

**Figure 5 Effect of garlic extracts on the membrane permeability (A) and protein leakage (B) of** *X. campestris* **pv.** *malvacearum.* **Values are means ± SD (*n* = 3).** (C) Control; (T) Taicangbaipi; (S) Single-clove; (E) Ershuizao; (H) Hongqixing.

## Changes in cell membrane permeability

Membrane permeability experiments were performed to study the effect of the garlic extracts on bacterial cell membranes. Figure 5A shows the effects of the extracts from the different garlic varieties on the membrane permeability of *X. campestris* pv. *malvacearum*. During the first 8 h, there was almost no change in the relative conductivity of the control; however, an increase in the conductivity was observed at 12 h, which may have been due to bacterial death. Similar to the control, the relative conductivity of the samples

for 'Single-clove' and 'Taicangbaipi' were only slightly higher. However, compared to the control, with increasing treatment time and concentration of the 'Hongqixing' and 'Ershuizao' extracts, the conductivity increased rapidly. This meant that the permeability of the bacterial membrane would have increased correspondingly, resulting in the loss of intracellular components, especially $K^+$, $Na^+$, and other electrolytes.

## Leakage of cellular contents

Similar to the membrane permeability test, the extracts from the different garlic varieties elevated protein leakage through the plasma membrane of *X. campestris* pv. *malvacearum* (Fig. 5B). 'Single-clove' and 'Taicangbaipi' caused slight protein leakage, whereas 'Hongqixing' and 'Ershuizao' caused extensive protein leakage. These results indicate that the 'Hongqixing' and 'Ershuizao' extracts may cause irreversible damage to the bacterial plasma membrane, resulting in the loss of cellular components such as proteins and some essential molecules, leading to cell death.

## Analysis of cytotoxic (antitumor) effects

After 72 h of incubation with extracts from different garlic varieties at different concentrations (dilution ratios of 0, 1/32, 1/16, 1/8, 1/4, and 1/2), we analysed the viability of SW480 (Fig. 6A) and HCT116 (Fig. 6B) cells. The cell viability for the control (no added garlic extract) was 100%. After incubation with the extracts from the different varieties of garlic at different concentrations, the cell viability decreased by varying degrees. At 1/2 concentration, after 72 h, the SW480 cell viability was 21.18% when incubated with 'Ershuizao', 29.14% with 'Hongqixing', 38.46% with 'Single-clove', and 39.07% with 'Taicangbaipi'. The extracts of 'Ershuizao' and 'Hongqixing' had strong cytotoxic effects on HCT116, with cell viabilities of 5.80% and 7.13%, respectively. Conversely, the cell viability for HCT116 with 'Single-clove' was 40.76% and with 'Taicangbaipi' was 40.88%.

## DISCUSSION

Four varieties of garlic are commercially available: 'Taicangbaipi', 'Ershuizao', 'Hongqixing', and 'Single-clove', and 'Ershuizao' and 'Hongqixing' are unique to Sichuan Province of China. In this study, we compared the components of the four garlic varieties, and found that the extracts of 'Hongqixing' and 'Ershuizao' exhibited significant antimicrobial and antitumor effects.

Table 1 shows the large variations in the levels of soluble sugar, total starch, and protein in the four varieties of garlic. Table 2 and Figs. 2 and 3 show that the constituents of the four garlic varieties also differed substantially. As shown in Table 2, 'Ershuizao' and 'Hongqixing' contained high levels of 5-hydroxymethylfurfural, at 47.10% and 26.78%, respectively. 5-Hydroxymethylfurfural is a common product of the Maillard reaction, which occurs during heat processing and the preparation of many types of foods and beverages (*Hwang et al., 2011*; *Molina-Calle, Priego-Capote & Castro, 2017*). 5-Hydroxymethylfurfural is common in black garlic, which is prepared by heat treatment (*Choi, Cha & Lee, 2014*; *Lu et al., 2017*); however, there have been no reports of this

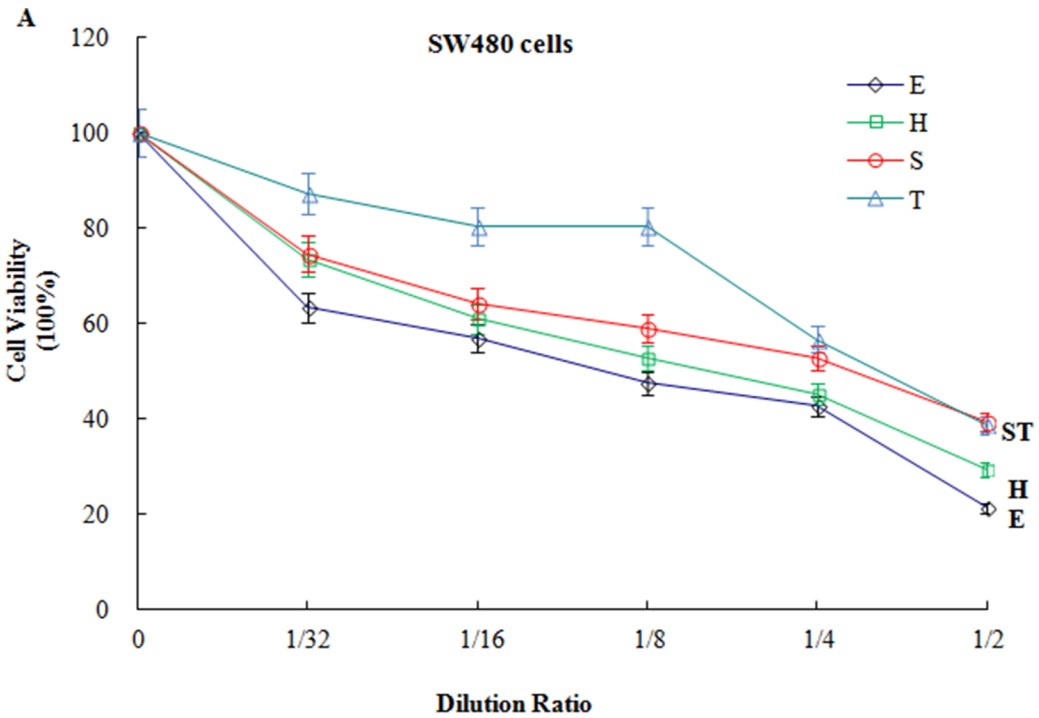

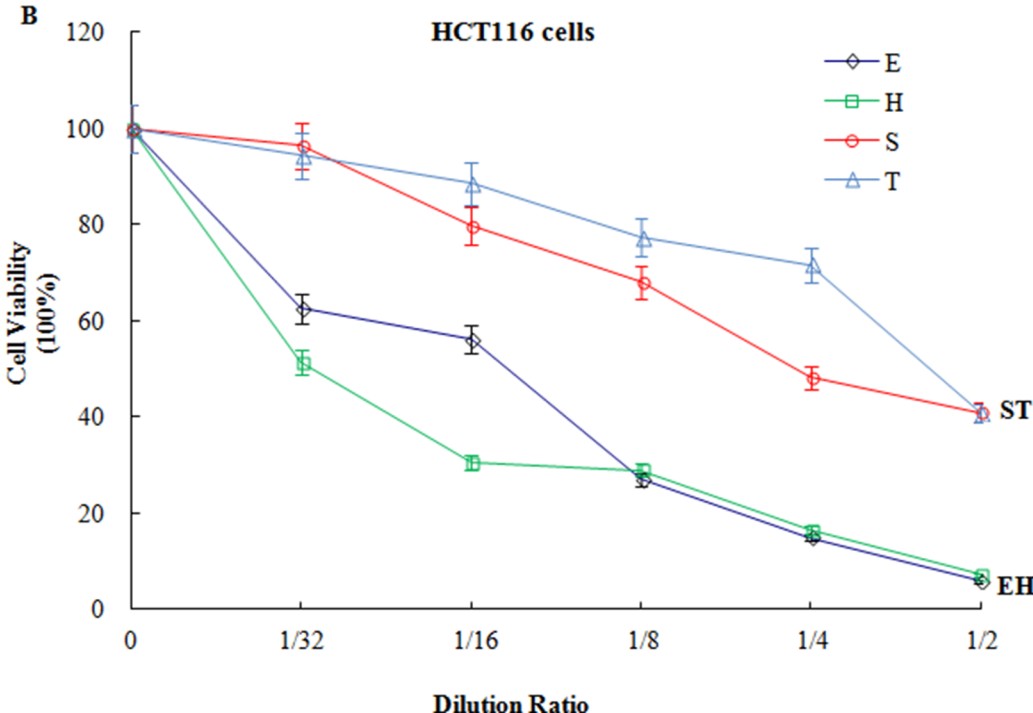

**Figure 6** **Analysis of cytotoxic effects on SW480 cells (A) and HCT116 cells (B).** Values are means ± SD ($n = 3$). (E) Ershuizao; (H) Hongqixing; (S) Single-clove; (T) Taicangbaipi.

compound in fresh garlic. Perez found that sucrose degrades into glucose and a very reactive fructofuranosyl cation under acidic conditions, and that this cation can be efficiently converted to 5-hydroxymethylfurfural (*Perez & Yaylayan, 2008*). 5-Hydroxymethylfurfural has many beneficial effects that have become increasingly apparent, including antitumor (*Michail et al., 2007*), antioxidant (*Li et al., 2009*), and cytoprotective (*Ding et al., 2010*) effects. Figure 6 show that both the 'Ershuizao' and 'Hongqixing' extracts, respectively, exhibited significant cytotoxic effects on CRC cells (SW480 and HCT116). These results suggest that under acidic conditions (pH 3.0), 'Ershuizao' and 'Hongqixing' garlic easily react and convert sucrose to 5-hydroxymethylfurfural, which has antitumor effects. However, previous studies on the antitumor effects of garlic focused on other organosulphur compounds, such as diallyl disulphide (DADS) (*Liao et al., 2009*; *Yin et al., 2018*), DATS (*Li & Lu, 2002*), and ajoene (*Li et al., 2002*).

In this study, we found low levels of some sulphides in the garlic extracts that could still be noteworthy (Figs. 2 and 3): 3-vinyl-1,2-dithiacyclohex-4-ene (0.98% in 'Hongqixing', 0.16% in 'Ershuizao'), 3-vinyl-1,2-dithiacyclohex-5-ene (1.14% in 'Taicangbaipi', 4.46% in 'Hongqixing', 1.48% in 'Ershuizao'), and DATS (0.74% in 'Taicangbaipi', 1.64% in 'Hongqixing', 0.38% in 'Single-clove'). Many studies have shown that garlic has an inhibitory effect on microorganisms (*Curtis et al., 2004*; *Mirik & Aysan, 2005*; *Slusarenko, Patel & Portz, 2008*), and the bacteriostatic effect of garlic is known to be related to its organic sulphur compounds (*Cavallito, Buck & Suter, 1944*; *Oommen et al., 2004*). The difference in the sizes of the inhibition zones in Fig. 4 may be associated with the presence of these sulphides. Many studies have shown that the composition of different species or varieties of plant extracts differs, and their inhibitory effects on bacteria also differ. *Khammuang & Sarnthima (2011)* reported differences in the antioxidant and antibacterial activities of four seed extracts from fresh Thai varieties of mango. Smolskaitė studied the antibacterial properties of wild mushroom extracts, and found that the extract of *Inonotus hispidus* was more effective against *Bacillus cereus*, *P. aeruginosa*, and *Candida albicans* than that of other mushrooms (*Smolskaitė, Venskutonis & Talou, 2015*).

The mechanism of how the garlic extracts kill microorganisms may be via the destruction of the bacterial plasma membrane. The observed changes in membrane conductivity and protein leakage indicate that the garlic extracts could destroy the structural integrity of the cell membranes of *X. campestris* pv. *malvacearum* (Fig. 5). Bacterial cell membranes provide conditions for the selective permeation of small ions such as $K^+$ and $Na^+$ (*Harold & Altendorf, 1974*; *Lanyi, 1979*), and once the cell membrane is damaged, loss of cell contents can lead to cell death (*Cui, Zhao & Lin, 2015*; *Ming et al., 2008*).

The garlic plant has a high value as a seasoning agent, and we found that the 'Hongqixing' and 'Ershuizao' varieties exhibit the best results in terms of their antimicrobial properties, 5-hydroxymethylfurfural production, and prominence of aromatic components. 'Hongqixing' and 'Ershuizao' are local garlic varieties, particular to the Sichuan Province of China; they are both quite famous for their strong aroma and storability. The present study has highlighted the value of the 'Hongqixing' and 'Ershuizao' varieties. Considering all the positive characteristics of the 'Hongqixing'

and 'Ershuizao' varieties, their production should be increased. Their antimicrobial properties could potentially be used in agriculture, for example, in organic farming, and their high production of 5-hydroxymethylfurfural could make them antioxidant health foods.

## CONCLUSIONS

In this study, four varieties of garlic were compared with respect to their nutrients, their levels of various compounds, antimicrobial and antitumor activity. Among these four varieties, 'Hongqixing' and 'Ershuizao' are unique to the Sichuan Province of China, and they are distinctive. Thus, we suggest that 'Hongqixing' and 'Ershuizao' should be given more prominent roles in agriculture and health foods.

### Funding

This work was financially supported by the Applied Basic Research Programs of Department of Science and Technology of Sichuan Province (No. 2017JY0178 and No. 2018JY0099) and Scientific Research Fund of Education Department of Sichuan Province (No.17ZB0071). The funders had no role in study design, data collection and analysis, decision to publish, or preparation of the manuscript.

### Grant Disclosures

The following grant information was disclosed by the authors:
Applied Basic Research Programs of Department of Science and Technology of Sichuan Province: 2017JY0178 and 2018JY0099.
Scientific Research Fund of Education Department of Sichuan Province: 17ZB0071.

### Competing Interests

The authors declare that they have no competing interests.

### Author Contributions

- Cun Chen conceived and designed the experiments, performed the experiments, analysed the data, prepared figures and/or tables, authored or reviewed drafts of the paper.
- Jing Cai performed the experiments, contributed reagents/materials/analysis tools.
- Song-qing Liu analysed the data, contributed reagents/materials/analysis tools.
- Guo-liang Qiu performed the experiments, analysed the data.
- Xiao-gang Wu analysed the data.
- Wei Zhang performed the experiments.
- Cheng Chen analysed the data.
- Wei-liang Qi analysed the data.
- Yong Wu analysed the data.
- Zhi-bin Liu conceived and designed the experiments, contributed reagents/materials/analysis tools, prepared figures and/or tables, authored or reviewed drafts of the paper, approved the final draft.

## Data Availability

The raw measurements are available in the Supplementary Files.

## Supplemental Information

Supplemental information for this article can be found online at http://dx.doi.org/10.7717/peerj.6442#supplemental-information.

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
