# Peer review of "Comparative study on the composition of four different varieties of garlic"

_PeerJ, doi:10.7717/peerj.6442_

## Round 0.1 · original submission · Major Revisions

Two reviwers have gone through your manuscript and returned their comments accordingly. Please carefully read these comments and suggestions for your revision, and make a list of point to point anwsers/explainations to each of reviewers' comments together with your revised manuscript. You probably need extra expenrimental data to satisfy the reviewers, especially testing more samples for validation of good sampling as suggested by reviewer1.

·

Basic reporting

Methods of comparison of samples are acceptable, with some details needs to be improved, such as using standards for identification compounds ore quantification of concentration of peaks.

Language is understandable

Format of Tables are ok, Raw are acceptable.

For conclusion of this article, results form only one batch samples are too weak. More data needs to be compared for samples from different resource (such as different growing area, different harvest season or storage condition samples....

Experimental design

Line from 84-90
Q: Why using 80% of ethanol for extraction of soluble sugar from garlic mashed tissue? Does this have reference of the method? My understanding of "soluble sugar" in plant is water soluble sugar.

Figure 2
Q: Suggest giving evidences of the m/z trial (mass spectrum) for each of the 7 identified peaks.

Validity of the findings

Line 173-184
Q: Identification of these compounds from GC-MS was based only from Nist library need more support evidences. Such as standards for the 7 key chemicals listed in the article . For quantification, internal standard or external standard (standard curve) should be required

Table 2
Q: Suggest using retention index for peak identification, rather than retention time.

Additional comments

For comparing difference of contents in biological samples, ripening stage, harvest time, storage time and temperature, growing area, soil pH et al…. are all the factors influenced in the testing results and needs to be considered. This article compared only one batch each of samples from supermarket . More experiments or evidences for different conditions or treatments of garlic samples are suggested to be compared for supporting the conclusion.

Suggestion: data from more batches of samples from different growing areas for each variety, or freshly harvest samples from similar storage conditions (similar storage time, temperature, et al) are supportable for this.

Reviewer 2 ·

Basic reporting

This paper explores difference of nutrients, compounds composition, antimicrobial and antitumor activity among four varieties of garlic. The authors claim to find that there are many distinctive differences between “Hongqixing” and “Ershuizao”, which are unique to Sichuan Province of China.

Experimental design

no comment

Validity of the findings

This is quite significant discovery and may improve the cultivation of garlic.

Additional comments

However, some slight points are missing in the manuscript, which are very important for a publication.
1. The author mentioned that “the garlic juices were mixed with deionized water at pH=3.0 at a ratio of 1.2 g/mL” in 2.5. It is not specific enough. The author should supplement the detailed conditions such as time and temperature.
2. Why did the author choose the two tumor cell lines SW480 and HCT116 to verify the antitumor activity of the extracts? A valid explanation should be given.
3. The author only tested the membrane permeability and leakage of protein in X. campestris pv. malvacearum. How about other microorganisms?
4. Why did the author choose deionized water at pH=3 to extract the garlic juices?
5. However, pH itself might influence the growth of microorganism. Therefore, the author should test the water control with pH=3.
6. Previous researches reported higher levels of sulphide components in garlic by GC-MS. Why is the content of sulphide components lower in this paper?

---

## Round 0.2 · accepted · Accept

Reviewer 2 is happy with your point to point answers in your revision. I have also gone through your responding to questions from Reviewer 1 who is not available for comments, and they seem to be resonable.

# Reviewer 2 ·

Basic reporting

no comment

Experimental design

no comment

Validity of the findings

no comment

Additional comments

I enjoyed reading the new version of the manuscript, which has addressed my concerned.
Therefore, I have no further comments to add.